# Deficiency of Caspase-1 Attenuates HIV-1-Associated Atherogenesis in Mice

**DOI:** 10.3390/ijms241612871

**Published:** 2023-08-17

**Authors:** Mohammad Afaque Alam, Maurizio Caocci, Mi Ren, Zheng Chen, Fengming Liu, Mst Shamima Khatun, Jay K. Kolls, Xuebin Qin, Tricia H. Burdo

**Affiliations:** 1Department of Comparative Pathology, Tulane National Primate Research Center, Tulane University School of Medicine, Tulane University, 18703 Three Rivers Road, Covington, LA 70433, USA; malam1@tulane.edu (M.A.A.); mrem@tulane.edu (M.R.); zchen29@tulane.edu (Z.C.); liufengming329@gmail.com (F.L.); 2Department of Microbiology and Immunology, School of Medicine, Tulane University, New Orleans, LA 70112, USA; 3Department of Microbiology, Immunology and Inflammation, Center for NeuroVirology and Gene Editing, Lewis Katz School of Medicine, Temple University, Philadelphia, PA 19140, USA; tuh20265@temple.edu; 4Departments of Pediatrics & Medicine, Center for Translational Research in Infection and Inflammation, Tulane University School of Medicine, New Orleans, LA 70112, USA; mkhatun@tulane.edu (M.S.K.); jkolls1@tulane.edu (J.K.K.); 5Department of Medicine, Section of Pulmonary Diseases, Critical Care and Environmental Medicine, Tulane University School of Medicine, New Orleans, LA 70112, USA

**Keywords:** HIV-1, atherosclerosis, macrophage, caspase-1, animal models

## Abstract

Within arterial plaque, HIV infection creates a state of inflammation and immune activation, triggering NLRP3/caspase-1 inflammasome, tissue damage, and monocyte/macrophage infiltration. Previously, we documented that caspase-1 activation in myeloid cells was linked with HIV-associated atherosclerosis in mice and people with HIV. Here, we mechanistically examined the direct effect of caspase-1 on HIV-associated atherosclerosis. Caspase-1-deficient (*Casp-1*^−/−^) mice were crossed with HIV-1 transgenic (Tg26^+/−^) mice with an atherogenic *ApoE*-deficient (*ApoE*^−/−^) background to create global caspase-1-deficient mice (*Tg26^+/^*^−^*/ApoE*^−/−^*/Casp-1*^−/−^*)*. Caspase-1-sufficient (*Tg26^+/^*^−^*/ApoE*^−/−^*/Casp-1^+/+^*) mice served as the controls. Next, we created chimeric hematopoietic cell-deficient mice by reconstituting irradiated *ApoE*^−/−^ mice with bone marrow cells transplanted from *Tg26^+/^*^−^*/ApoE*^−/−^*/Casp-1*^−/−^ (BMT *Casp-1*^−/−^) or *Tg26^+/^*^−^*/ApoE*^−/−^*/Casp-1^+/+^* (BMT *Casp-1^+/+^*) mice. Global caspase-1 knockout in mice suppressed plaque deposition in the thoracic aorta, serum IL-18 levels, and ex vivo foam cell formation. The deficiency of caspase-1 in hematopoietic cells resulted in reduced atherosclerotic plaque burden in the whole aorta and aortic root, which was associated with reduced macrophage infiltration. Transcriptomic analyses of peripheral mononuclear cells and splenocytes indicated that caspase-1 deficiency inhibited caspase-1 pathway-related genes. These results document the critical atherogenic role of caspase-1 in chronic HIV infection and highlight the implication of this pathway and peripheral immune activation in HIV-associated atherosclerosis.

## 1. Introduction

Cardiovascular disease (CVD), including atherosclerosis and atherosclerosis-associated complications, is an increasingly common comorbidity and cause of mortality in people with HIV (PWH) in the post-antiretroviral therapy (ART) era [1,2,3,4,5,6]. ART suppresses but does not eradicate human immunodeficiency virus (HIV) in PWH. PWH on ART, and even elite controllers (PWH who maintain undetectable viral loads for at least 12 months without starting ART), exhibit an increased incidence of CVD that includes atherosclerosis, stroke, and heart attacks [7,8]. HIV-1 primary target cells are CD4^+^ T-cells, binding to co-receptors such as CCR5 and CXCR4, and monocytes, where the virus activates a plethora of signaling mechanisms including NLRP3-mediated caspase-1 activation [7,9,10,11,12,13]. Peripheral immune cell activation, however, is not indicative of cardiovascular inflammation, implying that the pathogenesis of HIV-associated vascular inflammation may be different from peripheral inflammation [14,15,16,17]. However, the cellular and molecular mechanisms underlying HIV-1-accelerated atherogenesis remain to be fully defined [6,7]. Understanding its molecular mechanisms will aid in the design and development of novel interventions to treat HIV-associated CVD [6,7,18].

Over the past few decades, our team and other groups have studied the effect of HIV-associated diseases including atherosclerosis [6,7,19,20,21]. The Tg26 mouse model is one of the three HIV transgenic lines originally generated on an FVB genetic background by using a noninfectious HIV-1 DNA construct with a deletion of the 3 kb region of the *gag/pol* genes. The noninfectious Tg26 mouse, which has viral transcripts but no active viral replication, develops HIV comorbidities. This suggests that persistence of HIV transcripts and viral proteins alone without replication is sufficient to trigger inflammation and comorbidities [21,22]. To model HIV-associated atherogenesis, we crossed Tg26 mice with atherogenic *ApoE*-deficient mice (*Tg26^+/^*^−^*/ApoE*^−/−^). Using this model, we documented that HIV expression globally or locally in hematopoietic cells accelerated atherogenesis in these mice [23]. Additionally, HIV-1-accelerated atherogenesis is related to an elevated level of caspase-1 and monocyte/macrophage activation [23]. Here, we further examined this model to dissect the NLRP3/caspase-1 pathway in HIV-associated atherogenesis, which has not been fully elucidated. 

To address this, we crossed global caspase-1-deficient gene to *Tg26^+/^*^−^*/ApoE*^−/−^ (*Tg26^+/^*^−^*/ApoE*^−/−^*/Casp-1*^−/−^) and generated chimeric *ApoE*^−/−^ mice carrying the HIV transgene along with caspase-1 deficiency in hematopoietic cells via a bone marrow transplantation (BMT) approach. Here, we documented that global caspase-1 deficiency or specific caspase-1 deficiency in immune cells attenuates HIV-associated atherogenesis in mice. This is associated with reduced macrophage content in plaque lesions. Taken together, these results highlight the atherogenic role of caspase-1 in HIV-associated macrophage activation and atherosclerosis and indicate that therapeutic inhibition of caspase-1 may have a beneficial effect on treating HIV-associated atherosclerosis.

## 2. Results

### 2.1. Global Caspase-1 Knockout Decreases Plaque Size Only in the Thoracic Region of the Aorta in Tg26^+/−^/ApoE^−/−^/Casp-1^−/−^ Mice and Reduces Foam Cell Formation

Our previous studies demonstrated that caspase-1 activation in myeloid cells was associated with HIV-associated atherosclerosis in HIV transgenic (*Tg26^+/^*^−^*/ApoE*^−/−^) mice and people with HIV [19]. To determine the causative effect of caspase-1 in HIV-1-associated atherosclerosis, we compared the development of atherosclerosis in caspase-1-sufficient vs. -deficient mice carrying the HIV transgene with an atherogenic background (*Tg26^+/^*^−^*/ApoE*^−/−^*/Casp-1^+/+^* vs. *Tg26^+/^*^−^*/ApoE*^−/−^*/Casp-1*^−/−^). To facilitate atherogenesis, the mice were fed an atherogenic diet for 12 weeks. We found that the deficiency in caspase-1 in *Tg26^+/^*^−^*/ApoE*^−/−^*/Casp-1*^−/−^ mice resulted in significantly fewer atherosclerosis plaques in the thoracic aorta (Figure 1A,C), but not in the entire aorta including the arch, thoracic, and abdominal aorta (Figure 1A,B), nor in the aortic root (Figure 1D–F). We did not detect any changes in body weight, cholesterol levels, or levels of triglycerides between the groups (Figure 1G–I). 

Macrophage foam cell formation is a crucial step in the development of atherogenesis. Here, we investigated the role of caspase-1 in foam cell formation in HIV-accelerated atherosclerosis. Monocyte-derived macrophage (MDM) cells, isolated from the spleens of caspase-1-sufficient and -deficient mice, were cultured for eight days and incubated with/without oxLDL to examine the ex vivo formation of foam cells. We found that global caspase-1 knockout in *Tg26^+/^*^−^*/ApoE*^−/−^*/Casp-1*^−/−^ mice statistically reduced foam cells compared to *Tg26^+/^*^−^*/ApoE*^−/−^*/Casp-1^+/+^* mice (Figure 2A,B). NLPR3 inflammasome components and downstream cytokines were examined in MDMs obtained from the spleens of caspase-1-sufficient and -deficient mice before and after oxLDL treatment, and only mature IL-1β RNA levels were altered (Appendix A). We further measured the levels of the cytokine IL-18, which is downstream of caspase-1 activation in serum [23,24]. Consistently, our results showed a significant reduction in IL-18 in caspase-1-deficient mice compared to caspase-1-sufficient mice (Figure 2C). However, as expected, we did not observe differences in other cytokines (IL-2, IL-5, IL-6, IL-10, IL-1, and KC/GRO; Appendix A). Taken together, our results suggest that global caspase-1 deficiency attenuates the development of plaque lesion only in the thoracic aorta in *Tg26^+/^*^−^*/ApoE*^−/−^*/Casp-1*^−/−^ mice, which is associated with a reduced level of IL-18 and reduced foam cell formation.

### 2.2. Deficiency of Caspase-1 in Immune Cells Attenuates Atherogenesis in HIV Transgenic Mice

Previously, we used the BMT approach to generate chimeric *ApoE*^−/−^ mice expressing HIV transcripts in hematopoietic cells and demonstrated that the expression of HIV transcripts in hematopoietic cells accelerated atherogenesis [23]. Herein, to minimize the confounding effect of global caspase-1 deficiency and maximize the effect of caspase-1 deficiency in hematopoietic cells on plaque development, we performed BMT to generate chimeric *ApoE*^−/−^ mice reconstituted with bone marrow from either *Tg26^+/^*^−^*/ApoE*^−/−^*/Casp-1^+/+^* (BMT *Casp-1^+/+^)* or *Tg26^+/^*^−^*/ApoE*^−/−^*/Casp-1*^−/−^ (BMT *Casp-1*^−/−^*)* mice [23], which are more clinically relevant models than global HIV transgenic and caspase-1 knockout mice. We investigated HIV transcripts in several tissues using qPCR in these chimeric mice as well as *Tg26^+/^*^−^ mice. The chimeric mice carrying either *Tg26^+/^*^−^*/ApoE*^−/−^*/Casp-1^+/+^ or Tg26^+/^*^−^*/ApoE*^−/−^*/Casp-1*^−/−^ bone marrow expressed comparable levels of HIV transcripts in multiple immune tissues, including the spleen, thymus, and aorta, as *Tg26^+/^*^−^ mice (Figure 3A–C). Together, these results demonstrate the successful establishment of the HIV chimerism in these mice carrying caspase-1-deficient and -sufficient genes.

To explore the causative effect of caspase-1 on atherosclerosis in immune cells, the chimeric mice carrying HIV transcripts along with caspase-1-deficient or -sufficient genes in immune cells were fed an atherogenic diet for eight weeks. The chimeric mice deficient in caspase-1 had significantly less plaque burden in the aorta (Figure 3D–F) and aortic root (Figure 3G–I) than the chimeric mice sufficient in caspase-1. Moreover, total cholesterol, but not triglycerides or body weight, was significantly lower in the chimeric mice deficient in caspase-1 (Figure 3J–L). The slight decrease in cholesterol in caspase-1-deficient mice may partially contribute to the lower plaque burden, but this needs to be further investigated. Together, these results suggest that caspase-1 deficiency in immune cells attenuates the progression and development of HIV-associated atherogenesis.

### 2.3. Reduced Caspase-1 Activation Is Associated with Reduced Macrophage Content in the Plaque of Chimeric ApoE^−/−^ Mice Reconstituted with Tg26^+/−^/ApoE^−/−^/Casp-1^−/−^ Bone Marrow

We further investigated immune cell content in our chimeric *ApoE*^−/−^ mice reconstituted with caspase-1-deficient (*Tg26^+/^*^−^*/ApoE*^−/−^*/Casp-1*^−/−^) or -sufficient (*Tg26^+/^*^−^*/ApoE*^−/−^*/Casp-1^+/+^*) bone marrow. We examined caspase-1 activity in CD11b+, Ly6C+ (inflammatory), and Ly6G+ (classical) monocytes in the chimeric mice. As expected, we found a significant decrease in caspase-1 activity in the chimeric *ApoE*^−/−^ knockout mice carrying caspase-1-deficient bone marrow compared to the mice carrying caspase-1-sufficient bone marrow (Figure 4A–C). We investigated the presence of CD68 (macrophage), CD3 (T-cell), and caspase-1 expression in the aortic root of bone-marrow-transplanted mice. There was a significant reduction in the number of lesion-associated CD68+ macrophages (Figure 4D–F) and caspase-1+ cells (Figure 4J–L), but not CD3+ T cells (Figure 4G–I), in the mice carrying caspase-1-deficient bone marrow when compared to their counterparts. We did not observe differences in serum IL-2, IL-4, IL-5, IL-6, IL-10, IL-12, IL-1, KC/GRO, INF, or TNF in the chimeric mice reconstituted with caspase-1-sufficient or -deficient bone marrow cells. Together, these results provide evidence supporting the notion that caspase-1 activation is associated with reduced macrophage content in plaque in *Tg26^+/^*^−^*/ApoE*^−/−^*/Casp-1*^−/−^ mice. These results together with the results presented above further indicate that caspase-1 activation contributes to HIV-1-associated atherosclerosis via macrophage activation.

### 2.4. Transcriptomic and Pathway Changes in PBMCs and Spleens of Chimeric ApoE^−/−^ Mice Reconstituted with Caspase-1-Deficient (Tg26^+/−^/ApoE^−/−^/Casp-1^−/−^) vs. Caspase-1-Sufficient (Tg26^+/−^/ApoE^−/−^/Casp-1^+/+^) Bone Marrow

To further investigate the global transcriptomic changes underlying the activation of caspase-1 in HIV-associated atherogenesis, we performed bulk RNA analysis of PBMCs and spleens of chimeric *ApoE*^−/−^ mice that received bone marrow from caspase-1-deficient (*Tg26^+/^*^−^*/ApoE*^−/−^*/Casp-1*^−/−^) mice or from caspase-1-sufficient (*Tg26^+/^*^−^*/ApoE*^−/−^*/Casp-1^+/+^*) mice. More than 500 genes were differentially expressed in the caspase-1-deficient mouse group than in the control group based on a cutoff value of 0.02. The data are shown as volcano plots and heat maps (Figure 5A,B).

The chimeric *ApoE*^−/−^ mice that received bone marrow from the caspase-1-deficient mice had significantly reduced caspase-1 gene expression in both PBMCs and spleen, compared to those who received bone marrow from the caspase-1-sufficient mice, which further confirmed the successful reconstitution of bone marrow cells in the chimeric *ApoE*^−/−^ mice (Table 1). *Eps8l1, Slc15a2*, and *Calpain11* were downregulated in both PBMCs and spleens of the chimeric *ApoE*^−/−^ mice reconstituted with *Tg26^+/^*^−^*/ApoE*^−/−^*/Casp-1*^−/−^ bone marrow cells (Table 1). Additionally, some genes, such as *CD14, Myc, GRB2*, and *Aqp1*, were differentially expressed in either PBMCs or spleen and involved in caspase-1 pathway modulations [25,26,27,28] (Table 2).

We further conducted pathway analyses to demonstrate that several genes are differentially expressed with significant biological functions associated with caspase-1 deficiency in both PBMCs (Figure 6A,B) and spleen (Figure 6C,D). In PBMCs, we detected that the chimeric *ApoE*^−/−^ mice carrying *Casp*-1-deficient bone marrow had significantly increased activation of the hallmark_heme_metabolism pathway (Figure 6A), and reduced activation of the hallmark_TNFalpha_signaling_via_NFκ-beta, hallmark_hypoxia, and hallmark_inflammatory_response pathways (Figure 6B), than the chimeric *ApoE*^−/−^ mice carrying the *Casp*-*1*-sufficient gene. Interestingly, we also found that complement pathway activation had a tendency to be reduced in PBMCs of the chimeric mice carrying *Casp-1*-deficient bone marrow (*p* = 0.094) (Figure 6B). In the spleen, we found that the *ApoE*^−/−^ mice carrying *Casp-1*-deficient bone marrow had significantly increased activation of certain pathways, e.g., hallmark_estrogen_response, hallmark_epithelial_mesenchymal_transition, and hallmark_myc_targets_v1 (Figure 6C), and reduced activation of the hallmark_heme_metabolism pathway (Figure 6D). Of note, in PBMCs, reduced activation of hallmark TNF-α signaling via NF-κB was detected, along with hallmark inflammatory responses and complement pathways in the mice deficient in *Casp-1* activation in immune cells, supporting our hypothesis that the deficiency in *Casp-1* pathway activation in macrophages attenuates HIV-associated atherogenesis. Taken together, changes both at the transcriptomic level and at the pathway level further clarify the pathogenic function of the caspase-1 pathway in HIV-associated atherogenesis.

## 3. Discussion

Herein, we demonstrated the causative effect of caspase-1 activation in HIV-1-associated atherosclerosis using HIV-1 transgenic mice under an atherogenic background and diet. A global caspase-1 knockout in mice suppressed plaque deposition in the thoracic aorta, which is associated with a reduced level of IL-18 in the circulation and decreased ex vivo foam cell formation. To further demonstrate the specific effect of caspase-1 in immune cells, we developed and used a clinically relevant mouse model, a hematopoietic cell-specific caspase-1-deficient and HIV transgene mouse model with an *ApoE*^−/−^ atherogenic background. We discovered that the absence of caspase-1 in mouse immune cells resulted in significantly reduced plaque size in the entire aorta and aortic root. These results document that caspase-1 activation directly contributes to the development of HIV-associated atherosclerosis. These results are consistent with our previously published findings: (1) caspase-1+ macrophages present in aortic plaques of PWH and (2) elevation of IL-18 in the plasma of PWH correlated with the total segments and number of non-calcified inflammatory plaques [23]. Further, we demonstrated that SIV infection in rhesus macaques increased caspase-1 activation and secretion of IL-18 [24]. SIV infection activates the NF-κB pathway [24], a first and critical signal for transcriptional upregulation of NOD-like receptor protein 3 (NLRP3), an upstream signaling molecule of caspase-1 activation and pro-IL-1β [29,30,31]. In the *Tg26^+/^*^−^*ApoE*^−/−^ mouse model, we found that caspase-1 activation in inflammatory monocytes correlated with HIV-associated atherogenesis [23]. Together, the results reported here further highlight the important role of caspase-1 in HIV-associated atherosclerosis.

Previous studies have experimentally examined the atherogenic role of caspase-1 in a non-HIV setting using caspase-1 deficiency in two complementary mouse models, *ApoE*^−/−^ and *Ldlr*^−/−^ [32,33,34]. Usui et al. reported that *ApoE*^−/−^ caspase-1 knockout mice have reduced atherosclerotic lesion formation when fed an atherogenic diet for 12 weeks [32]. Similarly reduced atherosclerotic lesion formation was found by inducing caspase-1 deficiency in *ApoE*^−/−^ mice under both a normal chow diet for 26 weeks and an atherogenic diet for 8 weeks [33]. Conversely, Menu et al. used *ApoE*^−/−^*Nlrp3*^−/−^, *ApoE*^−/−^*Asc*^−/−^, and *ApoE*^−/−^*Casp1*^−/−^ double-knockout mice and detected no significant differences in plaque size or macrophage infiltration compared to ApoE^−/−^ mice fed an atherogenic diet for 11 weeks [34]. The dissimilar outcomes of the two studies can be explained by the use of distinctive mouse models [35]. The bone marrow transplantation approach was also used to create chimeric *ApoE*^−/−^ and *Ldlr*^−/−^ mice carrying caspase-1-deficient or -sufficient bone marrow and again showed reduced plaque formation [35,36]. Our results in an HIV-1 transgenic mouse model are consistent with previous findings observed in non-HIV chimeric mice carrying caspase-1-deficient bone marrow [35]. However, in the case of global capsase-1 deficiency, we only observed significant plaque reduction in the thoracic areas, but not in the entire aorta or aortic root. There was a greater variability in the plaque burden in the entire aorta and aortic root compared to the thoracic region, which may account for the results only reaching significance for the thoracic region.

In the current study, we used multiple approaches to document the causative effect of caspase-1 activation in promoting HIV-associated atherogenesis. We found that global caspase-1 deficiency reduced serum IL-18 and foam cell formation. We observed a significant reduction in monocyte/macrophage infiltration in the aortic root plaque in caspase-1-deficient hematopoietic chimeric mice. Overall, there were fewer CD3+ T cells in the aortic plaques compared to the number of CD68+ macrophages. In the blood, there were fewer inflammatory monocytes (Ly6C+) in caspase-1-deficient mice compared to caspase-1-sufficient bone-marrow-cell-transplanted mice. Ly6C+ inflammatory peripheral monocytes migrate to the intima. Once in the intima, monocytes differentiate into macrophages, which phagocytize oxLDL, leading to foam cell formation and eventually formation of plaques, the hallmark of atherosclerosis. In the caspase-1-deficient mice, there was reduced inflammatory monocytes, decreased number of macrophages in the aortic root, and an attenuated amount of plaque. These results are supported by previous studies, which suggested a remarkable reduction in atherosclerosis in IL-18-deficient animals [36,37]. In SIV-infected rhesus macaques on an atherogenic diet, IL-18 levels and CD68-positive macrophages were directly linked with atherosclerotic lesion progression [38]. In vulnerable human atherosclerotic plaques, infiltrated macrophages correlated with IL-18 levels [39,40]. Increased expression of cell adhesion molecules such as VCAM1 in damaged endothelial cells mediates the adhesion and infiltration of circulating monocytes into the subendothelial space [41]. Nevertheless, the causative role of caspase-1-mediated myeloid cell activation in HIV-associated atherogenesis warrants further investigation [42,43,44].

We have reported that transcriptomic analyses of peripheral mononuclear cells and splenocytes indicate that the deficiency of caspase-1 inhibits the function of caspase-1 pathway-related genes such as *Eps8l1*, *Slc15a2*, *Calpain11*, *Cd14*, *Myc*, *Grb2*, and *Aqp1*. EPS8l-1, an epidermal growth factor receptor pathway substrate 8 protein, has been documented to regulate caspase-1 activation pathways in the absence of NLRP3 inflammasomes [45]. *Slc15a2*, a multi-pass membrane protein expressed in macrophages, is activated by external stimuli and then enhances inflammatory cytokine production [46]. *Capn11* (*Calpain11*) participates in cholesterol metabolism and release of caspase-1 from macrophages [47]. Additionally, calpain 11 is implicated in inflammatory cytokines, mediators, immune cells, and signaling cascades depending on the external stimuli [48]. Together, the downregulation of *Eps8l1*, *Slc15a2*, and *Calpain11* in our experimental setting further indicate that deficiency of caspase-1 inhibits the function of these caspase-1 pathway-related gene functions, which may contribute to HIV-associated atherogenesis. These genes need further investigation regarding their possible role in HIV-associated atherogenesis. Additionally, we also found that some genes are expressed differentially in a tissue-specific manner. These genes include CD14, Myc, GRB2, and Aqp1, which are involved in caspase-1 pathway modulations [25,27,49,50] and are differentially regulated in PBMCs and spleens. Finally, we documented that *Casp-1* pathways in PBMCs may participate in the activation of several inflammatory pathways, including hallmark TNF-α signaling via NF-κB, hallmark inflammatory response pathways, and complement pathways (*p* = 0.09), which may contribute to HIV-associated atherogenesis. Indeed, these findings are consistent with the previous findings reported by us [51,52,53,54] and other researchers [55], showing that complement pathways and NF-κB activation contribute to atherogenesis and aneurysm, and inhibition of these pathways has a beneficial effect in treating atherogenesis. Taken together, changes at the transcriptomic level and pathway activation in our HIV-1 transgenic caspase-1-deficient mouse model provide further insight into the pathogenic role of caspase-1 alteration in HIV-1-associated atherogenesis. However, the cause–effect of these genes and the pathways’ activation in HIV-associated anthogenesis require further investigation.

## 4. Materials and Methods

### 4.1. Mouse Treatment, Bone Marrow Transplantation, and Atherosclerotic Plaque Analyses

The use of animals in this study was approved by the Animal Care and Use Committee (IACUC) at Tulane University School of Medicine, and all experiments conformed to the relevant regulatory standards. HIV *Tg26^+/^*^−^ transgenic *ApoE*-deficient mice with B6 background (*Tg26^+/^*^−^*/ApoE*^−/−^) were generated by crossing *Tg26^+/^*^−^ mice with B6 background with *ApoE*^−/−^ mice, as described previously [23]. *Tg26^+/^*^−^*/ApoE*^−/−^*/Casp-1*^−/−^ and *Tg26^+/^*^−^*/ApoE*^−/−^*/Casp-1^+/+^* mice were generated by crossing *Tg26^+/^*^−^*/ApoE*^−/−^ mice with caspase-1-deficient mice (*Casp-1*^−/−^). *Casp-1*^−/−^ mice were purchased from Jacksons’ laboratory (B6N.129S2-Casp1^tm1Flv^). PCR genotyping was used to identify casp-1^+/+^, casp-1^+/−^, and casp-1^−/−^ mice during crossing. All animals used in the global knockout studies were confirmed to have their appropriate genotype.

Eight-week-old recipient *ApoE*^−/−^ mice were irradiated (9.5 Gy) using an RS2000 irradiator (Rad Source, Buford, GA), followed by transplanting 10^6^ donor bone marrow cells from *Tg26^+/^*^−^*/ApoE*^−/−^*/Casp-1^+/+^* or *Tg26^+/^*^−^*/ApoE*^−/−^*/Casp-1*^−/−^ mice through tail vein injection within 5 h of isolation [56]. The reconstituted (chimeric) mice were fed a normal chow for 5 weeks to allow full reconstitution [23,44] and then switched to an atherogenic diet (D12108C; Research Diets, Inc., New Brunswick, NJ, USA) of 20.1% saturated fat, 1.37% cholesterol, and 0% sodium cholate for 8 weeks [23]. The mice with global deficiency were fed the same diet for 12 weeks. For the global deficiency studies, 22 *Casp-1*^−/−^ and 9 *Casp-1^+/+^* animals were used in the studies. For the bone marrow transplantation studies, 19 *Casp-1*^−/−^ and 10 *Casp-1^+/+^* animals were used in the studies. At the culmination of the experiments, the mice were perfused with isotonic saline and sacrificed. Hearts along with aortic trees were removed for morphometric analyses. For the bone marrow transplantation studies, 5 *Casp-1*^−/−^ and 5 *Casp-1^+/+^* aortas were used for HIV envelope (env) expression and 14 *Casp-1*^−/−^ and 5 *Casp-1^+/+^* aortas were used for plaque characterization and histology. Plaques were identified in the whole aorta and aortic root using Oil Red O and H&E staining, respectively, to show the structural and morphological characteristics within the plaque sample. A digital camera was used to capture the whole image, and the total number of pixels for the whole aortas and plaque areas were measured using Image J, thus allowing calculation of the percentage of the total surface area of plaque in the aortas. The plaque area in the aortic root for each section was quantified, as described previously [23]. The chimerism was established by determining HIV transcript levels in multiple tissues, including the spleen, thymus, and aorta, using qRT-PCR.

### 4.2. Serum Lipid Analysis

The mice were anesthetized using isoflurane and sacrificed via cervical dislocation. Blood was collected from the heart using a 1 mL syringe. After 30 min at room temperature to allow clotting, the tubes were centrifuged at 2000× *g* for 10 min and serum was collected in fresh tubes and stored for analysis. Serum cholesterol and triglyceride were measured at the central facility of Tulane University. For the global studies, 5 *Casp-1*^−/−^ and 2 *Casp-1^+/+^* animals did not have serum available for lipid analyses.

### 4.3. Atherosclerotic Plaque Analysis in Aorta

For atherosclerotic analysis, whole aortic trees were excised from the mice and placed in 4% PFA. After processing, the hearts were embedded in an optimal cutting temperature (OCT) compound. A total of 30 serial histological sections were obtained at a 7 μm thickness from the aortic valve toward the ascending aorta, as described previously [23]. Whole aortic trees were processed, followed by Oil Red O staining. Residual tissues and fats from the aorta were removed, which was then pinned on the gel. Images were taken for plaque lesion quantification.

### 4.4. Histological Analysis of Plaque Progression in Aortic Sinus

Histological slides were processed using hematoxylin II and eosin Y (Abcam, Cambridge, MA, USA) staining to determine plaque lesions in the aortic sinuses as per the manufacturer’s protocol (Abcam, USA). Images were captured using light microscopy at 10× magnification and analyzed for plaque deposition.

### 4.5. Immunohistochemistry

To establish the macrophage and T-cell content in the studied mouse plaques, serial frozen sections of the aortic root (7 μm) were washed gently 3 times for 3 min each with 1× PBS and then blocked with Bloxall blocking solution (Vector lab, Newark, CA, USA), before incubating with specific antibodies (Appendix A). The antibodies were used at optimal dilutions, and dilutions were made using Tris-buffered saline (TBS) containing 3% bovine serum albumin (BSA). Incubation occurred overnight at 4 °C. After rinsing gently, the sections were treated with the respective fluorescent-labeled or HRP-conjugated secondary antibodies. After washing, the slides were mounted with DAPI mounting media. Images were taken at 10× magnification.

Caspase-1 immunohistochemistry in aortic plaque: After the blocking step, the slides were incubated with a rabbit-anti-mouse casp-1 antibody (Santa Cruz, sc-1218R, Santa Cruz, CA, USA) in the blocking buffer overnight at 4 °C, followed by horseradish peroxidase (HRP)-conjugated goat-anti-rabbit-IgG secondary antibody (DAKO, P0448, Northolt, UK). After washing three times, the slides were developed using 3,3-diaminobenzidine (DAB) from vector labs (SK-4105). Finally, the slides were mounted after hematoxylin QS counterstain (Vector Labs, USA, H-3404-100), and images were taken under the microscope at 10× magnification to analyze the caspase-1 content in the plaque area.

CD3 immunohistochemistry for T-cells in the plaque: The slides were incubated overnight at 4 °C with anti-mouse-CD3 monoclonal antibody (Santa Cruz Biotechnology, Dallas, TX, USA), followed by fluorescent-labeled secondary antibody. The slides were mounted using DAPI mounting media, and images were taken using a Nikon fluorescent Inverted microscope (Model: Eclipse Ti2).

CD68 immunohistochemistry in resident macrophages: For macrophage staining, the slides were incubated using rat anti-mouse CD68 monoclonal antibody (Invitrogen, Waltham, MA, USA) overnight at 4 °C, followed by washing with PBS in tween 20, and then incubated with a secondary antibody for 1 h at room temperature. The slides were washed and mounted using DAPI mounting media. Images were captured using a fluorescent microscope.

### 4.6. FAM-FLICA Caspase-1 Staining and Flow Cytometry Analysis

Active caspase-1 activity in monocytes was determined using the FAM FLICA Casp-1 assay kit (Immunochemistry Technologies, Davis, CA, USA) (Appendix A), following the manufacturer’s instructions. In a subset of bone-marrow-transplanted animals, blood from *Tg26^+/^*^−^*/ApoE*^−/−^*/Casp-1^+/+^* (*n* = 4) and *Tg26^+/^*^−^*/ApoE*^−/−^*/Casp-1*^−/−^ mice (*n* = 4) was collected via tail bleeding. After removal of red blood cells, peripheral blood mononuclear cells (PBMCs) were incubated with monocyte markers, including CD11b-PE-Cy7, Ly6C-Flour450 (eBioscience, San Diego, CA), and Ly6G-APC (Bio legend, San Diego, CA, USA), for 30 min at room temperature (Appendix A). After washing with 1xPBS, the cells were then stained with caspase-1 inhibitor FAM-YVAD-FMK at 37 °C for 1 h, followed by washing with the 1x apoptosis wash buffer provided in the kit. In the end, the supernatants were discarded, and the cells were resuspended with 300 μL of 1× apoptosis wash buffer. Caspase-1 activity in monocytes was recorded using BD LSR Fortessa flow (Becton Dickinson Immunocytometry Systems, San Jose, CA, USA). The analysis and histogram graphics were performed using the FlowJo software (version FlowJo_v10.9.0_CL) (Ashland, OR, USA).

### 4.7. Foam Cell Formation Assay

Mouse spleens were harvested at Tulane University and sent to Temple University. A cell suspension was obtained using a gentleMACS Dissociator homogenizer, followed by filtration through a 40 µm cell strainer. Splenic monocytes were isolated from the cell suspension using an EasySep™ mouse monocyte isolation kit (STEMCELL Technologies Inc., Vancouver, BC, Canada) and plated. To differentiate into macrophages, the cells were treated every 2 days for 8 days with RPMI 1640 containing M-CSF at 50 ng/mL and 10% FBS (Appendix A). To induce foam cell formation, adherent macrophages were treated with 100 μg/mL of oxidized low-density lipoprotein (oxLDL), from Kalen Biomedical, Montgomery Village, MD, USA), for 24 h (Appendix A) and then double-stained with 0.4% Oil red O for detection of lipids and CD163 antibody for detection of tissue macrophages (Appendix A). The percentage of macrophages that became foam cells and were lipid positive was quantified in multiple fields. Spleens were not available for 3 global *Casp-1*^−/−^ animals.

### 4.8. qRT-PCR

According to the manufacturer’s instructions, total RNA was extracted from either cells (splenocytes) or tissues (thymus, spleen, and aorta) using a Monarch^®^ Total RNA Miniprep Kit (New England Biolabs, Ipswich, MA, USA). A total of 100 ng of RNA was used for each reaction using a Luna^®^ Universal Probe One-Step RT-qPCR Kit (New England Biolabs, Ipswich, MA, USA) and Roche light cycler 96 (Roche, Indianapolis, IN, USA). The reaction conditions were as follows: reverse transcription at 55 °C for 10 min, initial denaturation at 95 °C for 1 min, denaturation at 95 °C for 10 s, extension at 60 °C for 30 s (+plate read), for a total of 45 cycles. Data for relative expression compared to beta-actin were analyzed using the 2^−ΔΔCt^ method. The primers and probes utilized were as follows: NLRP3 Forward ACCTTTGCCCATACCTTCAG, NLRP3 Reverse TGCCACAAACCTTCCATCTA, NLRP3 Probe TCTTCCTGTTAACTGACCATCCCGC, IL-18 Forward TTGCTTTCACTTCTCCCCTG, IL-18 Reverse AGCATGGAACCACAGAGAAC, IL-18 Probe AGTCCAACTGCAGACTGGCACA, casp-1 Forward TTCAACATCTTTCTCCGAGGG, casp-1 Reverse CACCTCTTTCACCATCTCCAG, casp-1 Probe CCCAGATCCTCCAGCAGCAACTT, ASC Forward ATTGCCAGGGTCACAGAAG, ASC Reverse AGGATGGAACAAAGCTGAAGAG, ASC Probe TGGACGGAGTGCTGGATGCTT, IL-1β Forward TGGGCTGGACTGTTTCTAATG, IL-1β Reverse TTTCTTGTGACCCTGAGCG, and IL-1β Probe CCCCTGGAGATTGAGCTGTCTGC.

### 4.9. ELISA and Meso Scale Discovery (MSD) Assay

Mouse serum was used to perform all the analyses. Mouse IL-18 was purchased from MBL International Corporation (Woburn, MA, USA). V-PLEX Proinflammatory Panel 1 Mouse Kit was purchased from Meso Scale Diagnostics (Rockville, MD, USA) and includes IL-1β, IL-2, IL-5, IL-6, KC/GRO, and IL-10. The levels of cytokines were determined following the manufacturer’s instructions.

### 4.10. RNA Extraction, Library Preparation, Sequencing, and Analysis

A new cohort of bone-marrow-transplanted animals were used in the RNA seq studies. Thirteen recipient *ApoE*^−/−^ mice were irradiated as described above and reconstituted with donor bone marrow cells of either female *Tg26^+/^*^−^*/ApoE*^−/−^*/Casp-1^+/+^* (*n* = 8) or male *Tg26^+/^*^−^*/ApoE*^−/−^*/Casp-1*^−/−^ mice (*n* = 5). After eight weeks of atherogenic diet, PBMCs and spleen tissues were harvested. For bulk RNA seq analysis of PBMCs and spleens, total RNAs were isolated using a RNeasy mini kit (Qiagen, CA, USA). The extracted RNA samples underwent quality control (QC) assessment using the Agilent Bioanalyzer (Agilent, Santa Clara, CA, USA), and all RNA samples submitted for sequencing had an RNA Integrity Number (RIN) >8. The RNA library was prepared for sequencing using Illumina TruSeq Stranded mRNA sample prep kit as previously described [57,58]. After the cDNA library preparation was completed, it was pooled and sequenced using NextSeq 500/550 High Output Kit on an Illumina NextSeq 550 as per the SMART-Seq Stranded Kit’s protocol (Takara Bio USA, Inc., San Jose, CA, USA). The mapping errors and ambiguity were eliminated, followed by normalization of the read counts across all samples. All subsequent analyses and differential expressions were performed using DESeq, EdgeR, and Cuffdiff (Slug Genomics, UV Santa Cruz), and the graphs and volcano plots were produced by applying the R program.

### 4.11. Statistics

The data were analyzed using GraphPad Prism 9.0 software (La Jolla, CA, USA) and presented as mean ± standard error of the mean (SEM). The experiments were performed using at least two biologically distinct replicates. Outliers were analyzed using the ROUT method, and values 3-fold greater than the SEM were excluded. One-way ANOVA with Bonferroni correction was used for multiple comparisons. All data sets were tested for normality and lognormality using the following methods: D’agostino–Pearson omnibus normality test, Anderson–Darling test, Shapiro–Wilk test, and Kolmogorov–Smirnov normality test with Dallal–Wilkinson–Lilliefor *p*-value. In all cases, either the sample size was too small, or the data sets did not pass the normality and lognormality tests, so the Mann–Whitney test, a non-parametric test, was used. Significant differences were determined at *p <* 0.05.

## 5. Conclusions

The molecular pathways involved in accelerated HIV-associated atherosclerosis were explored using both global and hematopoietic cell deficiency of caspase-1 in an HIV transgenic mouse model with an atherogenic background. The results showed that caspase-1 deficiency led to a significant reduction in atherosclerotic plaque in the aorta and decreased foam cell formation, thus attenuating HIV-associated atherogenesis in mice. Additionally, the bulk RNAseq showed changes at the transcriptomic level and pathway activation in caspase-1-deficient mice. These data further highlight the pathogenic role of the caspase-1 pathway in HIV-associated atherogenesis.

## Figures and Tables

**Figure 1 ijms-24-12871-f001:**
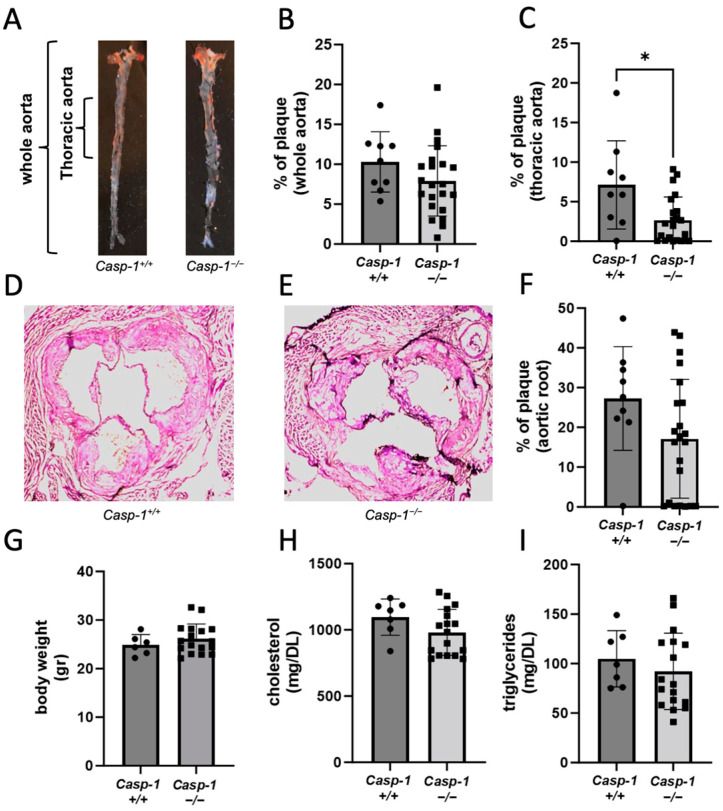
**Global caspase-1 knockout in *Tg26^+/^*^−^*/ApoE*^−/−^*/Casp-1*^−/−^ mice on 12 weeks of atherogenic diet decreases the percentage of plaques.** (**A**) Representative images of Oil red O staining of *Tg26^+/^*^−^*/ApoE*^−/−^*/Casp-1^+/+^ (Casp-1^+/+^)* and *Tg26^+/^*^−^*/ApoE*^−/−^*/Casp-1*^−/−^
*(Casp-1*^−/−^) aortas and quantification of the (**B**) whole aorta and (**C**) thoracic aorta sections (*Casp-1^+/+^*; n = 9), (*Casp-1*^−/−^; *n* = 22). (**D**,**E**) Representative images of H&E staining of the aortic root taken at 10× magnification and (**F**) quantification of plaque area. (**G**) Representative graphs of body weight (gr = grams) (*Casp-1^+/+^*; *n* = 6) (*Casp-1*^−/−^; *n* = 17), (**H**) serum cholesterol, and (**I**) serum triglycerides (*Casp-1^+/+^*; *n* = 7) (*Casp-1*^−/−^; *n* = 17). Two-tailed *t*-tests were used to compare *Casp-1^+/+^* vs. *Casp-1*^−/−^, * *p* < 0.05. Mann–Whitney non-parametric tests were used. Data are expressed as mean ± SEM.

**Figure 2 ijms-24-12871-f002:**
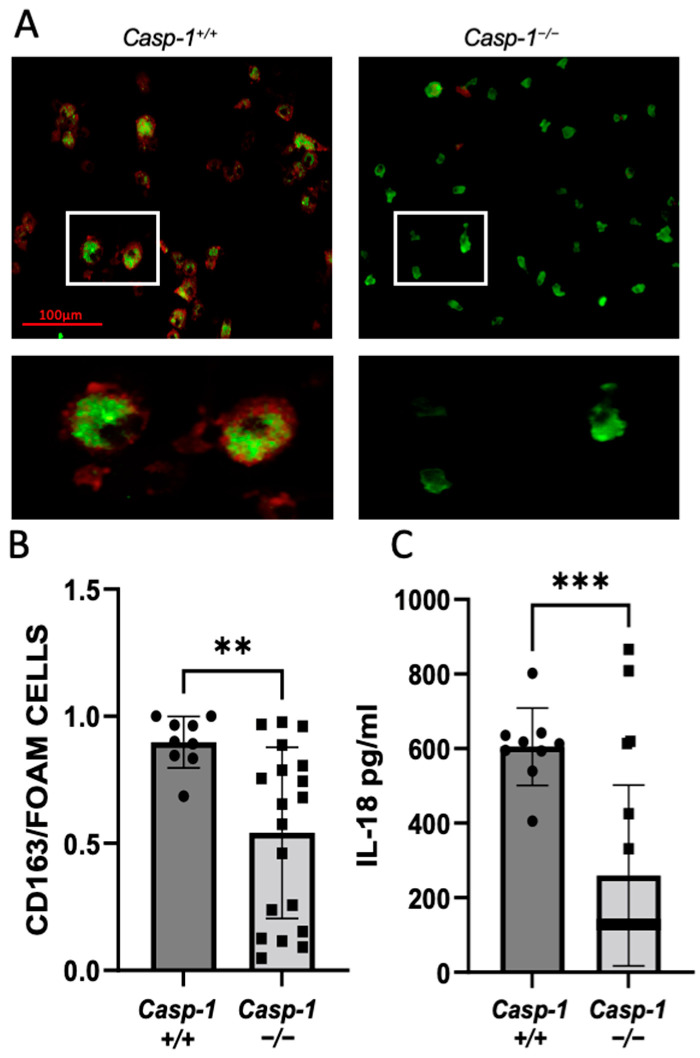
**Caspase-1 knockout in *Tg26^+/^*^−^*/ApoE*^−/−^*/Casp-1*^−/−^ mice on 12 weeks of atherogenic diet decreases foam cell formation and IL-18.** (**A**) Representative images taken at 20× magnification of Oil red O staining of monocyte-derived macrophages (MDMs) isolated from the spleens of caspase-1-sufficient (*Casp-1^+/+^)* and caspase-1-deficient (*Casp-1*^−/−^*)* mice, cultured for 8 days, and incubated without oxLDL (100 μg/mL) for 24 h. Cells were stained with Oil red O (red) for lipids and CD163 (green) for MDMs. Scale bar = 100 μM. (**B**) Quantification of CD163+ foam cells. Two-tailed t-test was used to compare *Tg26^+/^*^−^*/ApoE*^−/−^*/Casp-1^+/+^* mice *(Casp-1^+/+^*; *n* = 9) vs. *Tg26^+/^*^−^*/ApoE*^−/−^*/Casp-1*^−/−^ mice (*Casp-1*^−/−^; *n* = 19 (3 spleens were not viable when culturing)), ** *p* < 0.01. (**C**) IL-18 was measured using ELISA in serum from *Casp-1^+/+^* (*n* = 9) and *Casp-1*^−/−^ mice (*n* = 22) on 12 weeks of atherogenic diet. Data were analyzed using a two-tailed *t*-test, *** *p* < 0.001. Mann–Whitney non-parametric tests were used. Data are shown as mean ± SEM.

**Figure 3 ijms-24-12871-f003:**
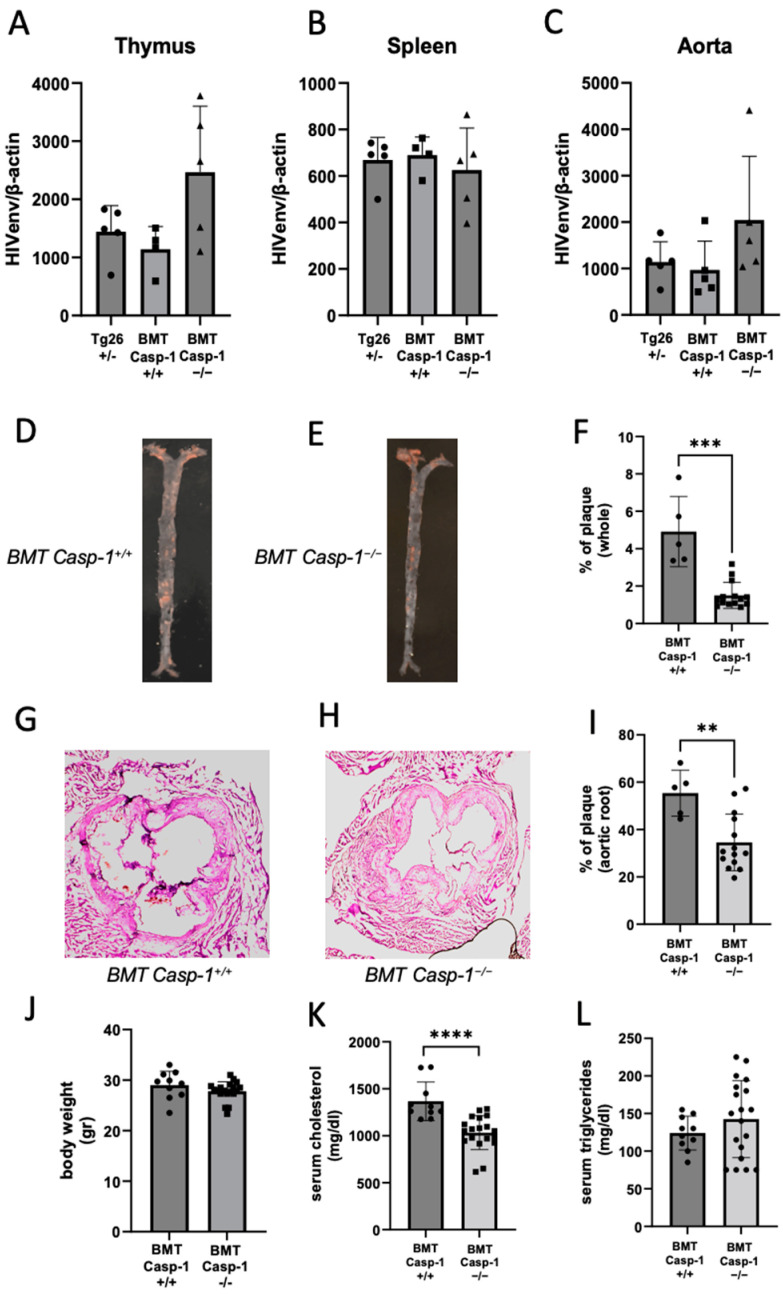
**Chimeric *ApoE*^−/−^ mice reconstituted with *Tg26^+/^*^−^*/ApoE*^−/−^*/Casp-1*^−/−^ bone marrow cells (BMT *Casp-1*^−/−^*)* have significantly less plaque formation than chimeric *ApoE*^−/−^ mice reconstituted with *Tg26^+/^*^−^*/ApoE*^−/−^*/Casp-1^+/+^* bone marrow cells (BMT *Casp-1^+/+^*) after 8 weeks of atherogenic diet**. HIV env transcript levels in the (**A**) thymus, (**B**) spleen, and (**C**) aorta of *Tg26^+/^*^−^*/ApoE*^−/−^ mice (*Tg26+/−*; *n =* 5), bone-marrow-transplanted *Tg26^+/^*^−^*/ApoE*^−/−^*/Casp-1^+/+^* mice (BMT *Casp-1^+/+^*; *n =* 5), and *Tg26^+/^*^−^*/ApoE*^−/−^*/Casp-1*^−/−^ mice (BMT *Casp-1*^−/−^; *n =* 5). Representative images of atherosclerosis using Oil red O staining of the whole aorta from (**D**) *Casp-1^+/+^* and (**E***) Casp-1*^−/−^ mice, and (**F**) quantification in *Casp-1^+/+^* (*n =* 5) and Casp-*1*^−/−^ mice (*n =* 14). Representative images of H&E staining of aortic root taken at 10× magnification from (**G**) *Casp-1^+/+^* and (**H**) *Casp-1*^−/−^ mice, and (**I**) quantification in *Casp-1^+/+^* (*n =* 5) and *Casp-1*^−/−^ mice (*n =* 14). In all animals used for evaluating HIV env transcripts and for aorta staining, (**J**) body weight (gr = grams), (**K**) serum cholesterol, and (**L**) serum triglycerides were measured in *Casp-1^+/+^* (*n =* 10) and *Casp-1*^−/−^ mice (*n =* 19). Mann–Whitney non-parametric tests were used to compare *Casp-1^+/+^* vs. *Casp-1*^−/−^. ** *p <* 0.01, *** *p <* 0.001, and **** *p <* 0.0001. Data are shown as mean ± SEM.

**Figure 4 ijms-24-12871-f004:**
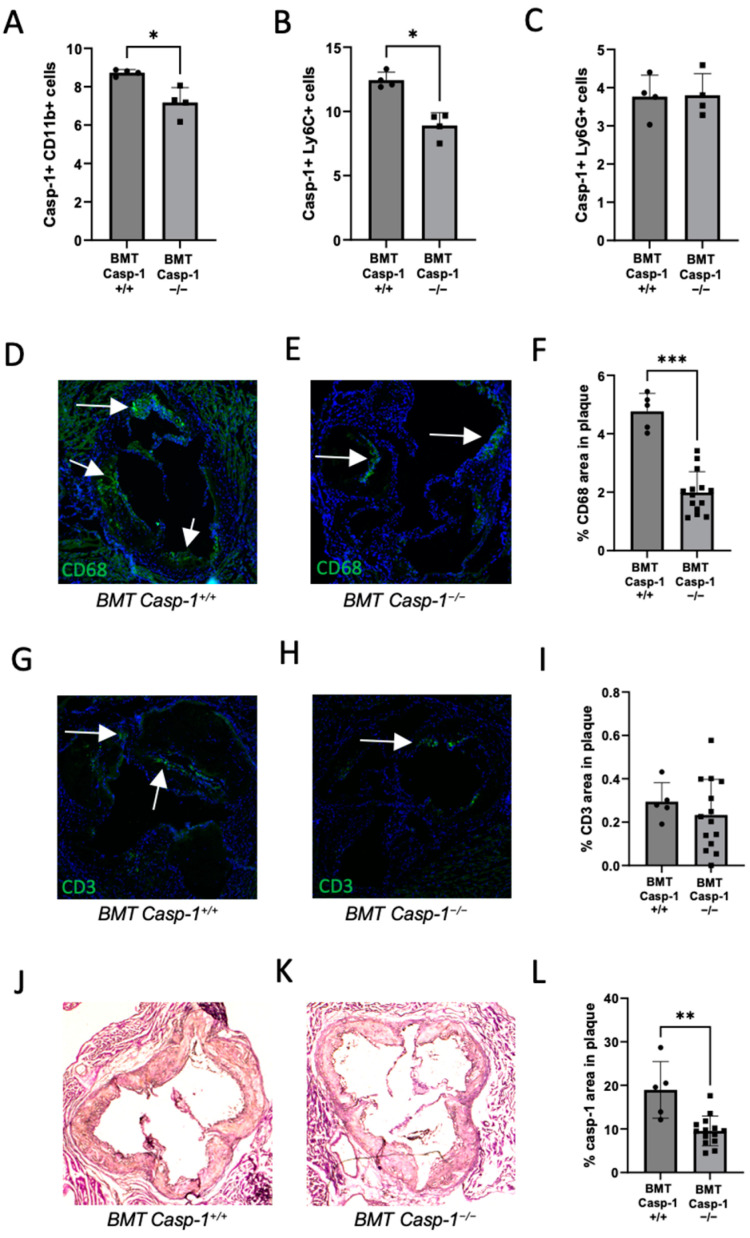
**Chimeric *ApoE*^−/−^ mice reconstituted with *Tg26^+/^*^−^*/ApoE*^−/−^*/Casp-1*^−/−^ bone marrow cells (BMT *Casp-1*^−/−^) have significantly less CD68 content and casp-1 activation and expression than chimeric *ApoE*^−/−^ mice reconstituted with *Tg26^+/^*^−^*/ApoE*^−/−^*/Casp-1^+/+^* bone marrow cells (*BMT Casp-1^+/+^*) after 8 weeks of atherogenic diet**. Caspase-1 activation in blood monocytes as detected by flow cytometry staining of (**A**) CD11b+, (**B**) Ly6C+, and (**C**) Ly6G+ in *Tg26^+/^*^−^*/ApoE*^−/−^*/Casp-1^+/+^* mice (BMT Casp-1^+/+^; *n =* 4) vs. *Tg26^+/^*^−^*/ApoE*^−/−^*/Casp-1*^−/−^ (BMT *Casp-1*^−/−^; *n =* 4) mice. CD68 staining in the aortas of (**D**) *Casp-1^+/+^* and (**E**) *Casp-1*^−/−^ mice and (**F**) quantification. CD3 staining in the aortas of (**G**) *Casp-1^+/+^* and (**H***) Casp-1*^−/−^ mice and (**I**) quantification. (**J**–**L**) Casp-1 staining in the aortas with quantification. For CD68, CD3, and caspase-1 staining, *Casp-1^+/+^* (*n =* 5) and *Casp-1*^−/−^ mice (*n =* 14) were used. Mann–Whitney non-parametric tests were used to compare *Casp-1^+/+^* vs. *Casp-1*^−/−^ mice. * *p* < 0.05, ** *p* < 0.01, *** *p* < 0.001. The white arrows are showing the positive staining for CD68 or CD3. Data are shown as mean ± SEM. Images were taken at 10× magnification.

**Figure 5 ijms-24-12871-f005:**
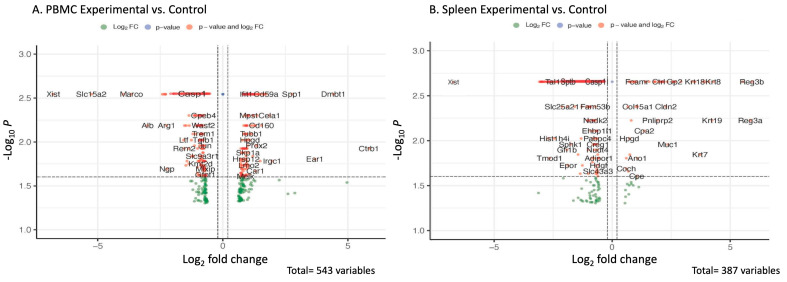
**Differentially expressed genes in PBMCs and spleens of chimeric *ApoE*^−/−^ mice reconstituted with *Tg26^+/^*^−^*/ApoE*^−/−^*/Casp-1*^−/−^ vs. *Tg26^+/^*^−^*/ApoE*^−/−^*/Casp-1^+/+^* bone marrow**. Volcano plot showing the most significantly expressed genes found via univariate analysis (cutoff value of 0.02). The plot presents significantly differentially expressed genes in the caspase-1-deficient (*Tg26^+/^*^−^*/ApoE*^−/−^*/Casp-1*^−/−^) compared to the caspase-1-sufficient mice (*Tg26^+/^*^−^*/ApoE*^−/−^*/Casp-1^+/+^*) in (**A**) peripheral blood mononuclear cells (PBMCs) and (**B**) spleen. The scatter plot of the negative log10-transformed *p*-values plotted against the log2 fold change. Negative values indicate downregulated genes in caspase-1-deficient mice (*Tg26^+/^*^−^*/ApoE*^−/−^*/Casp-1*^−/−^*),* while positive values reflect upregulated genes in caspase-1-deficient mice (*Tg26^+/^*^−^*/ApoE*^−/−^*/Casp-1*^−/−^*).* Genes with large fold-change values lie far from the vertical threshold line at log2 fold change = 0, indicating whether the genes are up- or downregulated.

**Figure 6 ijms-24-12871-f006:**
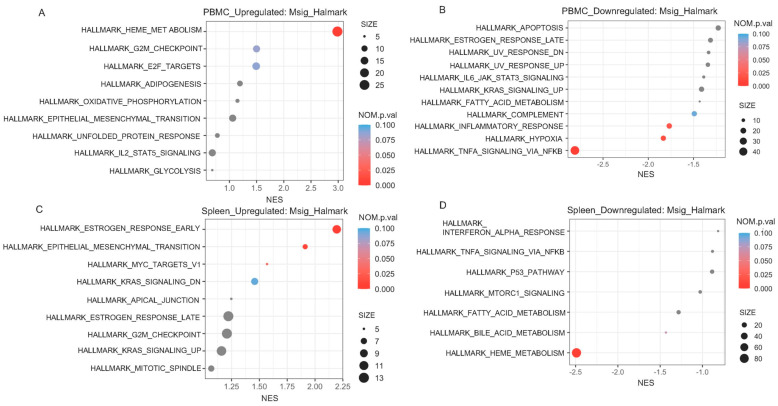
**Significant pathways based on PBMC and spleen data.** Enrichment plot of highly enriched biological pathways from the Molecular Signatures Hallmark gene set collection of upregulated genes and downregulated genes for (**A**,**B**) PBMCs and (**C**,**D**) spleen. The plot (**A**) presents only HALLMARK_HEME_METABOLISM as a significant pathway. The plot (**B**) shows HALLMARK_TNFA_SIGNALING_VIA_NFKB, HALLMARK_HYPOXIA, and HALLMARK_INFLAMMATORY_RESPONSE as significant pathways. The plot (**C**) shows 3 significant pathways, i.e., HALLMARK_ESTROGEN_RESPONSE_EARLY, HALLMARK_EPITHELIAL_MESENCHYMAL_TRANSITION, and HALLMARK_MYC_TARGETS_V1. The panel (**D**) shows HALLMARK_HEME_METABOLISM as a significant pathway. Significances in the panels were determined using Fisher’s exact test. NES = normalized enrichment score, NOM.p.val = normalized *p*-value.

**Table 1 ijms-24-12871-t001:** Differentially expressed genes observed in both PBMCs and spleens of chimeric *ApoE*^−/−^ mice reconstituted with *Tg26^+/^*^−^*/ApoE*^−/−^*/Casp-1*^−/−^ vs. *Tg26^+/^*^−^*/ApoE*^−/−^*/Casp-1^+/+^* bone marrow cells.

Gene	Chromosome Location	Fold Change	*p*-Value
Casp-1	Chr-9	−1.36	0.002
Eps8I1	Chr-7	−4.93	1.04 × 10^−18^
Slc15a2	Chr-16	−5.42	3.37 × 10^−40^
Capn11	Chr-17	−6.15	7.72 × 10^−6^

Abbreviations: Casp-1: Caspase-1; Capn11: calpain 11; EPS8l1: EPS8-like 1; Clc15a2: solute carrier family 15 (H+/peptide transporter), member 2.

**Table 2 ijms-24-12871-t002:** Differentially expressed genes observed in either PBMCs or spleens of chimeric *ApoE*^−/−^ mice reconstituted with *Tg26^+/^*^−^*/ApoE*^−/−^*/Casp-1*^−/−^ vs. *Tg26^+/^*^−^*/ApoE*^−/−^*/Casp-1^+/+^* bone marrow cells.

Gene	Chromosome Location	Fold Change (PBMC)	Fold Change (Spleen)	*p* Value
GRB2	Chr-11	1.87		0.004
Aqp1	Chr-6	2.05		0.036
Ppp3cc	Chr-8	−0.89		0.024
CD14	Chr-18		−1.50	0.004
Myc	Chr-11		−1.55	0.013
Ear1	Chr-14		3.94	6.31 × 10^−9^

Abbreviations: Aqp1: aquaporin 1; Ppp3cc: protein phosphatase 3 catalytic subunit gamma isoform; CD14: CD14 antigen; Myc: myelocytomatosis oncogene; Ear1: eosinophil-associated ribonuclease A family, member 1.

## Data Availability

The authors will make all materials used to conduct this research available to other researchers upon reasonable request.

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
