# Peer review of "Deficiency of Caspase-1 Attenuates HIV-1-Associated Atherogenesis in Mice"

_ijms, 2023, doi:10.3390/ijms241612871_

Round 1
Reviewer 1 Report
Using knockout animals, the authors have proved the contribution of caspase-1, primarily expressed in immune cells, to HIV-associated atherogenesis.
The authors conclude that therapeutic inhibition of caspase-1 may have a beneficial effect on treating HIV-associated atherosclerosis.
In fact, this work is a continuation of an earlier study (ref.23) by the same group, where the role of caspase-1 in the development of atherosclerosis was investigated by other approaches.
Comments.
1. The groups of animals were not comparable in number (e.g., 9 vs 22: 6 vs 17). Why is that so? Is there a statistical justification for the sample size?
2. The distribution of the data cannot be described as normal. Nonparametric criteria should be used for the data comparison.
3. The description of the immunohistochemistry method requires clarification.
The authors write:
All slides were washed 3 times for 3 minutes each with 1X PBS and then blocked with Bloxall blocking solution (Vector lab), before incubating in specific antibodies (Table S1).
Is it blocking endogenous peroxidase?
Next:
Caspase-1 immunohistochemistry for aortic plaque: After blocking step, slides were incubated with a rabbit-anti-mouse casp-1 antibody (Santa Cruz, Dallas, USA) in the blocking buffer for overnight at 4°C followed by respective secondary antibody, followed by rabbit anti-goat-IgG in 10% normal mouse serum. Which antibody is a "respective secondary antibody"? Why the rabbit anti-goat-IgG was added is not entirely clear.
Slides were incubated for 1 hr at room temperature using horseradish peroxidase (HRP) conjugated goat-anti-HRP-IgG secondary antibody (DAKO, CA). Why was the additional staining with peroxidase-labeled goat anti-goat-IgG antibodies applied? For the purpose of signal amplification?
Table S1 does not list all "secondary" antibodies.
4. Why CD163 antibody was used for foam cell typing? CD163 is traditionally considered a marker for M2 cells. Was this antibody conjugated with a fluorescent dye?
5. How was a cell suspension obtained from spleens for subsequent isolation of monocytes?
6. What primers were used for qRT-PCR? No sequences are given.
7. In the microscopic pictures, the magnification is not given anywhere, nor is the microscope used to examine the tissue sections indicated.
8. Title of Figure S1. NRLP3 inflammasome pathway activation in spleens from Tg26+/-/ApoE-/-/Casp-1+/+ and Tg26+/-/ApoE-/-/Casp-1-/- mice after 12 weeks of an atherogenic diet. As follows from the text, the expression of NRLP3 inflammasome pathway components was studied in isolated cell cultures, not in the spleen.
9. The authors write "monocyte-derived macrophages (MDM) cells isolated from spleens of caspase-1 sufficient and deficient mice, were cultured for 8 days, and incubated with/without oxLDL to examine the ex vivo formation of foam cells”. In the caption to Figure 2 (A) it is written that the cells were incubated without oxLDL (100 μg/mL) for 24 hours. Where are the data with oxLDL?
10. Figure 4 (J, K). The authors report a significant decrease in the number of caspase-1+ cells in Casp-1-/- sections. However, the lesions shown have almost the same brightness of staining, with a slightly different hue. I can't see the differences in the photo, the magnification is very low. No hematoxylin staining of the slices is visible.
11. Fig 6. The font is not readable.
12. There are errors, e.g. anti-mice antibodies, fluorescence microscope. Errors in punctuation.
Reviewer 2 Report
In this original article by Mohammad Afaque Alam and colleagues, the authors documented decrease in caspase-1 reduce HIV-1-associated atherogenesis in mice. The methodical approach from global to chimeric and presentation of the data are praiseworthy. Here are my concerns.
1. In Fig1, a validation of Caspase-1 knockout is needed to start the story. Of course, the authors proved and claimed previously in a separate publication but even in the supplemented figure of RT-qPCR the caspase-1 knockout is not more than 30% in global caspase-1 knockout model. A immunoblot could answer it more clearly. It is also needed to mention which kind of samples has been used for RT-qPCR (line 365-370 have not mention of it).
Nonetheless, if checked carefully at least two of the extreme outliers from Fig1 B, C, and F indicate the knocking out might not be uniform for all animals which underestimate the significance of the plaque formation.
2. In Fig1 H and I the data presented as pg/mL. In mice, most of the references found are in mg/DL. If someone try to compare the values from Fig1 to other references, it did not match for WT or CTRLs. Interestingly, Fig3 K and L also contain the same experiments but this time as mg/dL and the values are exactly similar to pg/mL that is not possible. The authors should recheck their values and need to represent with same units for Y-axis and keep up with the consistency.
3. In Fig2, the authors investigated the Interleukins. Fig2C showed the high variability in n=22 of cap-1 -/-, spanning from ~100 to 800 pg/mL while p value showed extreme significane. Interestingly, in supplements Fig S2 contains IL-2, 5, 6, 10, 1 beta and the n is exact opposite for casp1+/+ vs casp1-/-. These also need to be rechecked.
4. In Fig3, it is not clear from the figures that this are done in chimeric mice. Especially for Fig3 D to H needs clarification with a better naming that could differentiate it from Fig1 A to F. Otherwise the question arises why 1D,E showed opposite results than 3G,H if they are same?
5. In Fig5, the scale could be rearranged for a better view of the spread. X-axis could be from -5 to +5 and Y-axis could go up to 3. The fonts are barely visible.
6. Fig6 definitely need bigger font size.
7. In line 374, in the methods authors reported gamma-interferon that is not found in the figures.
8. I could not find a title says “conclusion” in this manuscript.
9. Overall, the story has a merit but some data need intense attention before publication especially the values showed in biochemistry.
Round 2
Reviewer 1 Report
The authors took into account many of my comments and finalised the article.
The only point not considered is the statistical processing of the data. The authors write "If the data were not normally distributed, then a non-parametric test was used", but they still use only parametric criteria.
What criterion to test for normality of distribution was used? I doubt that at N=4 or 5 the data can be treated parametrically, at least you can't use a t-test to find statistically significant differences. The Mann-Whitney test is usually used in such cases.
Author Response
All data sets were tested for normality and lognormality using the following methods: D’agostino-Pearson omnibus normality test, Anderson-Darling test, Shapiro-Wilk test and Kolmogorov-Smirnov normality test with Dallal-Wilkinson-Lilliefor P value. In all cases either the sample size was too small, or the data sets did not pass the normality and lognormality tests so a Mann-Whitney, a non-parametric test, was used. This has been adjusted in the materials and methods and added to the figure legends.
Reviewer 2 Report
I have no more comments.
Author Response
Thank you so much.